# Design and Manufacture of Polarization-Independent 3D SOI Vertical Optical Coupler

**DOI:** 10.3390/mi14061268

**Published:** 2023-06-18

**Authors:** Shengtao Yu, Xiaoyu Li, Chengqun Gui

**Affiliations:** 1School of Power and Mechanical Engineering, Wuhan University, Wuhan 430072, China; yushengtao@whu.edu.cn; 2The Institute of Technological Sciences, Wuhan University, Wuhan 430072, China; xiaoyul@whu.edu.cn

**Keywords:** 3D, polarization independent, SOI, vertical optical coupler, laser-direct-writing lithography

## Abstract

An optical coupler is a key input/output (I/O) device in a photonic integrated circuit (PIC), which plays the role of light-source import and modulated light output. In this research, a vertical optical coupler consisting of a concave mirror and a half-cone edge taper was designed. We optimized the structure of mirror curvature and taper through finite-difference-time-domain (FDTD) and ZEMAX simulation to achieve mode matching between SMF (single-mode fiber) and the optical coupler. The device was fabricated via laser-direct-writing 3D lithography, dry etching and deposition on a 3.5 µm silicon-on-insulator (SOI) platform. The test results show that the overall loss of the coupler and its connected waveguide at 1550 nm was 1.11 dB in transverse-electric (TE) mode and 2.25 dB in transverse-magnetic (TM) mode.

## 1. Introduction

As the core device of the next-generation photoelectric transceiver module, the photon-integrated chip has advantages in power consumption, integration, cost and other aspects and gradually developed into the application of photon computing, biosensing and other fields. Effectively coupling optical signals from external fibers to waveguides of optical chips has always been one of the key research directions in the field of optoelectronic hybrid integration and optical packaging [1,2,3].

Although the active or passive waveguide devices based on the metastructure and the metasurface can effectively realize the mode separation or conversion of optical signals on the chip, SSC plays an indispensable role in the input and output of optical signals on the optical chip I/O port [4,5,6]. The main function of SSC is to realize the input of optical signals with different polarization states and reduce the insertion loss at the port. The mainstream SSC mainly has two categories: end coupler (EC) and grating coupler (GC). According to the current PDK provided by various foundries, the edge coupler has a typical insertion loss of about 1.5 dB [7,8,9]. At the same time, because the edge coupler can be coaxially coupled with fiber or FA (fiber array), it is easier to obtain a small-size package, which is used as a more mainstream SSC solution in high-speed transceiver modules. In recent years, the ECs based on a metasurface-grating structure have also achieved small insertion losses; because of the small feature size that this type of coupler has, it has a high requirement for the minimum machining line width of the lithography equipment, and thus it has not been applied in large-scale industrialization. The end face of EC is located on the side wall of the edge of the chip, so it is difficult to test the optical parameters by means of the end coupling for wafer-level testing before slicing. The optical coupling test of the optical fiber and optical chip in the vertical direction is mainly realized by GC. Since the GCs produced in general are composed of waveguide grating and the waveguide connected behind it, the coupling efficiency is mainly related to the grating period, the diffraction angle and the polarization state of the light source. For GCs, the typical insertion loss is about 1.7–3.2 dB. Low coupling efficiency and strong polarization correlation are the main factors restricting the performance of grating couplers. In the process of optical coupling, it is often necessary to design corresponding grating structures for TE/TM-mode light sources, or to realize polarization separation by designing two-dimensional bidirectional grating [10,11]. However, considering the application requirements of package coupling, it is difficult to realize the low difference coupling between the external light source and the optical chip in the vertical direction of two polarization states by using the single GC without changing the coupling position and coupling angle.

In this paper, a vertical optical coupler for matching a single-mode fiber to a waveguide at the 1550 nm band was designed. The vertical edge coupler was composed of a concave mirror and half-cone structure, equipped with the advantages of vertical coupling and low insertion loss. The device was fabricated on a 3.5 µm device layer of SOI through laser-direct-writing 3D lithography, inductively coupled plasma (ICP) dry-etching and film-coating. A schematic diagram of the vertical optical coupler is shown in Figure 1a; it consists of a half-cone coupler connected to the waveguide and a concave mirror. The curvature radius of the concave mirror is 35 µm, and the end face of the coupler is a half-cone structure with a radius of 3.5 µm. The function of the concave mirror is mainly used to change the transmission light path and the molding of the mode spot. The half-cone coupler could accept the mode spots reflected by the horizontal concave lens, and the design of the semi-conical structure on the thick-silicon layer further increases the coupling tolerance. The type of laser-direct-writing machine used to realize the 3D lithography is PM100 from 4PICO, which has a single spot-scanning speed of 300 mm/s. Considering the grayscale photoresist in the lithography process and the anisotropy of Si in the etching process, we chose the SOI substrate with a device layer of 3.5 µm for device manufacturing to ensure the line-width change in the process of micromachining. The SOI of the silicon device layer with a thickness of more than 2 µm is commercially available in research institutions, such as VTT [12] and CEA-LETI [10,13].

## 2. Design and Simulation of 3D SOI Vertical Coupler

Before the fabrication of the device, the optical simulation was carried out by using Lumerical FDTD software, and the 3D structure was designed and optimized. The half-cone coupler was gradually reduced from a semicircle with a curvature radius of 3.5 μm to a strip waveguide with a cross-sectional area of 2 μm × 1 μm. The length of the half-cone coupler is 55 μm. We set the light source as a TE/TM fundamental-mode Gaussian light source with a wavelength of 1550 nm and MFD of 6 µm to couple it with the end face of the half-cone coupler. The simulation result is shown in Figure 2a. By setting monitors in each section of the half-cone coupler, we found that the electric field distribution of both the TE and TM fundamental modes show a semicircular morphology during transmission. It can be concluded that, different from the invert edge coupler, which amplifies the pattern spots through the tip waveguide, the half-cone coupler is a type of adiabatic device, and the mode profile distribution of it is mainly determined by the external profile of its section. In contrast, the widely used inverted-tapered-edge coupler achieves slab amplification and low insertion loss through tip waveguides and cladding devices, and the end-face size of the cladding is basically the same as that of the single-mode fiber.

The concave mirror is designed by using ZEMAX software, and the curvature radius of the concave mirror is optimized. As shown in Figure 2b, when the horizontal distance between the mirror and the end face of the half-cone EC is set to 10 µm and the mirror curvature radius is 35 µm, the light reflected by the mirror forms an elliptical spot at the end of the edge coupler. The size of the spot is about 6.2 µm × 3 µm, which basically matches the optical mode field of half-cone coupler simulated by FDTD.

The coupling efficiency of the half-cone coupler at different wavelengths is also simulated by using FDTD. Figure 3 shows that the coupling efficiency is highest at 1550 nm and is higher than 76.5% (1.16 dB) over the entire simulation wavelength range (1450–1650 nm).

The coupling efficiency of the 3D coupler was simulated by using the sweep function of FDTD. As shown in Figure 4a,b, the coupling efficiency of the half-cone coupler is the highest when in the TE polarized mode, which is 84.5%, and it is 83.5% in the TM po-larized mode. For the designed 3D half-cone coupler, the simulation results show that although there is a certain mismatch between SMF with MFD of 6 µm and the section size of the edge coupler, the design of the 3D half-conical structure enables the vertical edge coupler to achieve a high coupling efficiency and a large coupling tolerance at the wavelength of 1550 nm. The 3 dB tolerance of the half-cone coupler in the horizontal direction is at least greater than 9 µm (±4.5 µm), and the 3 dB (50.12%) tolerance in the vertical direction is at least greater than 5 µm, as shown in Figure 4.

## 3. Micro-Fabrication of 3D SOI Vertical Coupler

As mentioned above, referring to the thick silicon process of the VTT and CEA-LETI platform, and considering the compatibility with the CMOS process, we chose an SOI with a 3.5 μm device layer to manufacture the device. The completed vertical optical coupler manufacturing process is shown in Figure 5a. First, a grayscale photoresist PR1 for laser-direct-writing lithography was spin-coated on the surface of the SOI wafer, and the photoresist was baked at 150 °C. Then, a grayscale image file corresponding to the 3D object was imported into the PM100 lithography machine to complete the 3D exposure. Similar to the 3D-printing equipment, the machine mainly realizes 3D processing through the grayscale map layout in BMP format. In the process of lithography, the light source distributes the exposure power by recognizing the grayscale value of each pixel in the image, so as to realize the 3D exposure of grayscale photoresist. The design of the gray layout directly determines the quality of device manufacturing. Figure 5b is the grayscale layout file that was designed by us. After exposure, the wafer was rapidly developed with MF-319 developer solution. After hot-baking, a 3D photoresist sacrificial layer was obtained. In the next step, by setting the appropriate gas ratio and power, we used ICP equipment to etch the SOI with the 3D photoresist sacrificial layer and over-etched it to the silica layer. The device structure of the half-cone coupler and concave mirror was obtained on the device layer. The SEM image of the device layer after etching is shown in Figure 1b. In order to further increase the reflection efficiency of the concave mirror, the Ti/Ag metal layer was selectively plated on the surface of the concave mirror through one-step exposure and metal-sputtering. Finally, a layer of SiO_2_ was deposited on the surface of the device through the plasma-enhanced chemical vapor deposition (PECVD) process as the cladding layer of the device. The optical microscope image of the manufactured 3D vertical optical coupler is shown in Figure 5c.

## 4. Testing and Analysis

The coupling efficiency of the manufactured 3D vertical optical coupler was tested. As shown in Figure 6, we assembled a set of optical coupling test systems, including a fiber-coupled laser at a wavelength of 1550 nm, single-mode fiber patch cable, polarization controller, large-area detector and several fixed fixtures. First, the light emitted from the laser is adjusted by a polarization controller. Then, the transmitted light passes through a single-mode fiber and exits at the tip of the tapered bare fiber. The MFD of the tapered fiber is 6 µm. During the dynamic coupling process, the tapered fiber always maintains a fixed inclination angle with the mirror, and the end face of the taper fiber should be as close to the concave mirror as possible. In order to reduce reflection and stray light at the end face of the half-cone coupler during coupling, a refractive index matching liquids with a refractive index of 1.43 was used to wrap the end face of the edge coupler, the concave mirror and the tip of SMF. The output power on one side of the waveguide was received by a large-area detector, and the ratio of the output power obtained from the waveguide was the coupling efficiency that we focused on.

Coupling tolerance tests are mainly performed on horizontal planes parallel to the SOI substrate. In this plane, the conical fiber moves in the radial and parallel directions of the half-cone edge coupler, which we define as the X and Y directions, respectively. The overall coupling efficiency of the coupler and its connected waveguide is obtained by dynamic coupling, which is 77.5% and 59.5%, respectively, in the TE and TM mode, and the corresponding loss is 1.11 dB and 2.25 dB. Although there is a gap between the actual test loss and the simulation structure, high coupling efficiency and polarization insensitivity are achieved by using a single device without changing the angle of the coupler. As shown in Figure 7c,d, the measured coupling efficiency was normalized to obtain the tolerance distribution. The 3 dB (50.12%) coupling tolerance of the vertical coupler was about 3.0 µm (±1.5 µm), which was significantly different from that of the half-cone coupler. According to the coupling theory, in addition to the misalignment between the light source and the energy center of the edge coupler, the relative angle between the axial of tapered fiber and the edge coupler will also greatly affect the coupling efficiency. The concave mirror obviously increases the angle change in the dynamic coupling process, which may be the reason why the 3 dB coupling tolerance of the vertical optical coupler is relative.

## 5. Conclusions

The 3D vertical coupler manufactured using laser-direct-writing lithography and the MEMS process showed high coupling efficiency for both TE and TM light sources. Although the maximum coupling efficiency of the coupler for the TM light source was somewhat different from the design value, the coupler realized polarization-insensitive optical coupling in the vertical direction. The 3 dB coupling tolerance is slightly lower than that of the grating coupler. Later, we plan to further improve the coupling efficiency and coupling alignment tolerance of the coupler by further optimizing the curvature of the mirror. To sum up, the abovementioned characteristics of the coupler give it considerable application potential in wafer-level testing, co-packaged optics (CPO) and 3D optical packaging.

## Figures and Tables

**Figure 1 micromachines-14-01268-f001:**
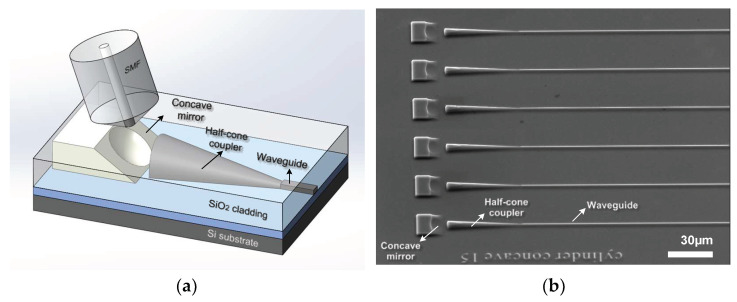
Demonstrated fiber coupling structure: (**a**) schematic diagram of vertical optical coupler and (**b**) 45-degree view of the SEM image after ICP etching.

**Figure 2 micromachines-14-01268-f002:**
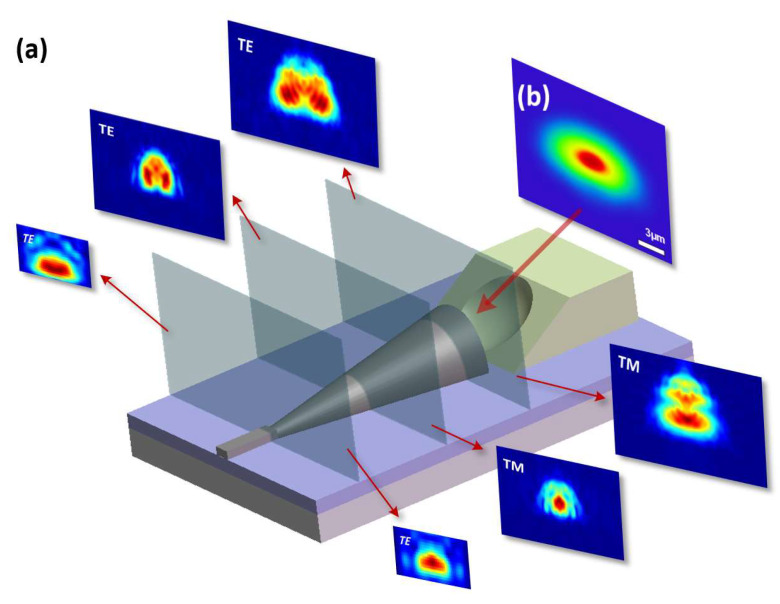
Simulation of the designed 3D vertical optical coupler: (**a**) mode field distributions of half-cone coupler; (**b**) mode field distributions of incident light field reflected by concave mirror.

**Figure 3 micromachines-14-01268-f003:**
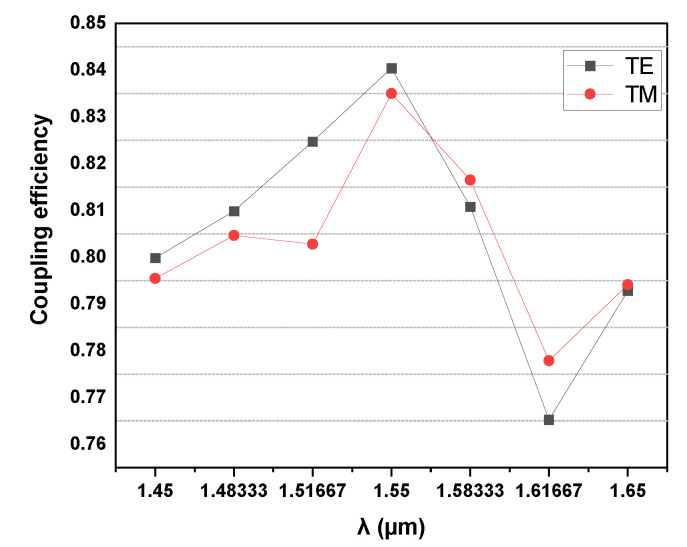
Coupling efficiency of 3D half-cone coupler corresponding to different wavelengths.

**Figure 4 micromachines-14-01268-f004:**
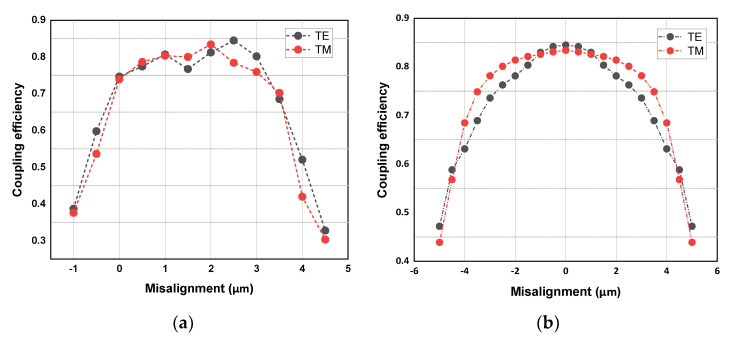
Simulation of the misalignment tolerance of vertical optical coupler: (**a**) alignment tolerance in vertical direction and (**b**) alignment tolerance in horizontal direction.

**Figure 5 micromachines-14-01268-f005:**
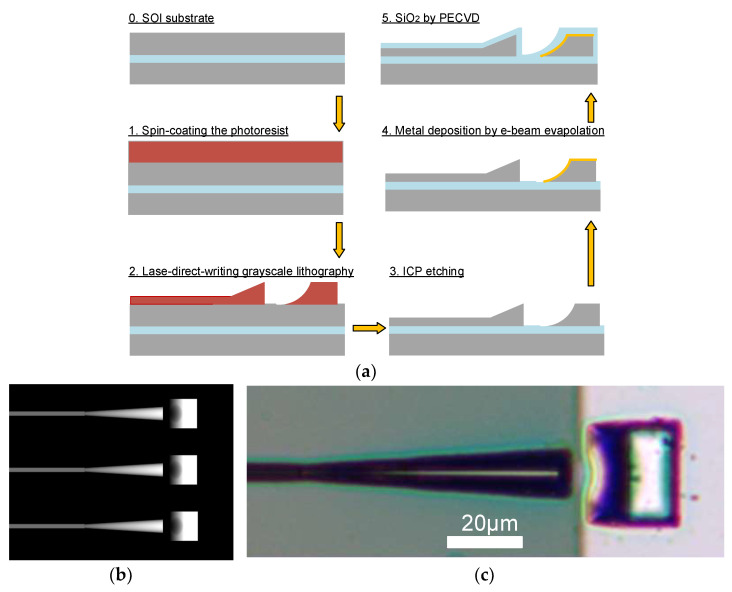
Micromachining steps of vertical optical edge coupler: (**a**) the process flow diagram, (**b**) gray layout image, and (**c**) a view of the object under optical microscope.

**Figure 6 micromachines-14-01268-f006:**
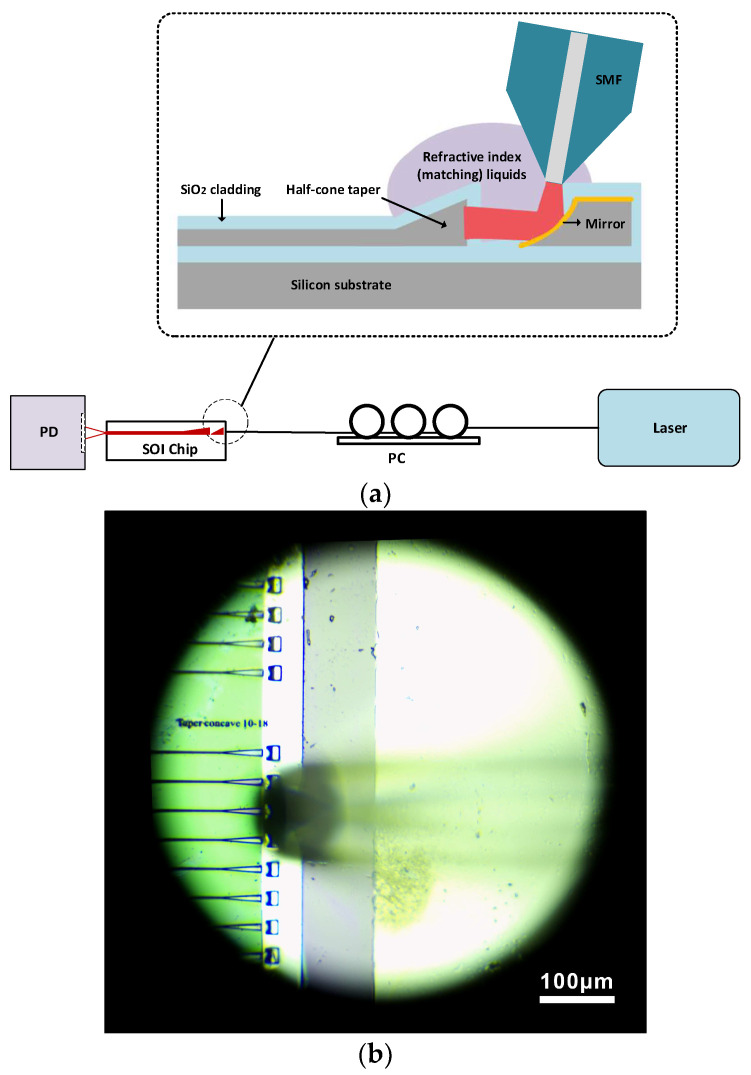
Testing system of 3D vertical optical coupler: (**a**) schematic diagram of optical fiber coupling test system and (**b**) top-view micro image of CCD.

**Figure 7 micromachines-14-01268-f007:**
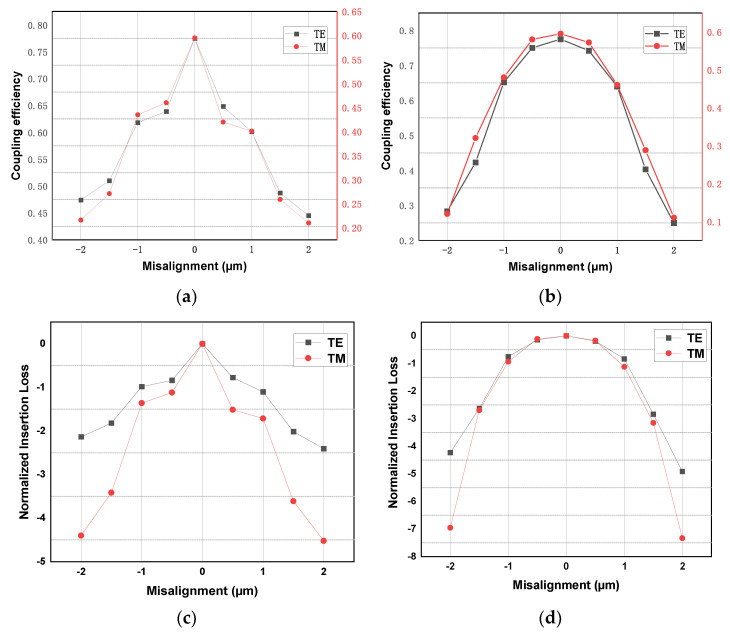
Misalignment tolerances of vertical optical coupling: (**a**) the coupling efficiency in the X direction; (**b**) the coupling efficiency in the Y direction, (**c**) normalized insertion loss in the X direction, and (**d**) normalized insertion loss in the Y direction.

## Data Availability

The data that support the findings of this study are available from the corresponding authors upon request.

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
