# Peer review of "Design and Manufacture of Polarization-Independent 3D SOI Vertical Optical Coupler"

_micromachines, 2023, doi:10.3390/mi14061268_

Round 1

Reviewer 1 Report

The authors reported a vertical coupler design for standard single mode fiber (SMF) and SOI waveguide with moderate coupling loss. FDTD simulations and corresponding experiments are performed to validate the polarization-independent coupler design.

The results are organized in a logical manner with agreement between design and experiments. However, before recommending publication of this manuscript, the authors should properly address the following technical comments to further enhance the manuscript.

1. As this paper is focused on the demonstration of on-chip couplers, at the beginning of the Introduction section, the authors should elaborate on the quest for integrated couplers and their significance/current issues, instead of Si- or other materials-based on-chip lasers that are less closely related to the core content.

2. Another point to address would be further highlighting manuscript novelty. From the second paragraph of the introduction, the pros and cons of conventional on-chip couplers are discussed. However, the authors did not mention an important category of integrated couplers made from subwavelength meta-structures (Ref: Optics Express, 22, 27175-27182, 2014. Ref: Light: Science & Applications 10, 235, 2021. Ref: Photonics Research 8, 564-576, 2020. Ref: Nature Nanotechnology, 12, 675–683, 2017. Ref: Scientific Reports, 8, 13362, 2018).

The authors need to benchmark with these references mentioned above to further highlight the advantages of the proposed device.

3. The resolution of the inset electric field distribution panels of Fig. 2 was low and better to be updated.

4. The optical bandwidth of the coupler is an important parameter. It will be helpful if the authors can add the FDTD simulation, or experimental data of the proposed device for the plot of coupling efficiency as a function of light wavelength.

5. In photonic integrated circuits, the lack of reconfigurability is one of the important issue. The authors can add comments on this regard on further potentially extending their designs to tunable devices by using phase-change materials (Ref: Nature Photonics, 11, 465–476, 2017. DOI: 10.1038/NPHOTON.2017.126) or van der Waals materials (Nature Reviews Materials, 2023. DOI: 10.1038/s41578-023-00558-w) to the revised manuscript.

The English writing is overall good.

Reviewer 2 Report

The Authors propose a 3D vertical optical coupler, acting as Spot Size Converter. The reported results have been experimentally achieved. The manuscript is well written and organized. Here, my comments:

1. The literature on the SSC should be improved, by citing several review papers on that topic (see, i.e., Mu, X., Wu, S., Cheng, L., & Fu, H. Y. (2020). Edge couplers in silicon photonic integrated circuits: A review. Applied Sciences10(4), 1538.)

2. The Authors report that the final waveguide is a strip one with a cross-section of 2 um x 1 um. However, these features make the waveguide not single mode. The excitation of further modes could be also responsible of the increasing of losses. Please clarify.

3. The approach to estimate the losses of the SSC is blurry. In particular, the Authors should label the losses related to the waveguide. Therefore, the use of a cut-back method is needed. 

4. Figure 5. The Authors use a 3 pads controller. However, the control of the exact polarization state is not accurate.

Round 2

Reviewer 2 Report

The Authors have modified the manuscript according to the Reviewer suggestions.

Author Response

Thank you for your support